# Invasive Fungal Diseases in Hospitalized Patients with COVID-19 in Israel: A Multicenter Cohort Study

**DOI:** 10.3390/jof8070721

**Published:** 2022-07-09

**Authors:** Meital Elbaz, Maya Korem, Oshrat Ayalon, Yonit Wiener-Well, Yael Shachor-Meyouhas, Regev Cohen, Jihad Bishara, Alaa Atamna, Tal Brosh-Nissimov, Nir Maaravi, Lior Nesher, Bibiana Chazan, Sharon Reisfeld, Oren Zimhony, Michal Chowers, Yasmin Maor, Eugene Katchman, Ronen Ben-Ami

**Affiliations:** 1Tel Aviv Sourasky Medical Center, Tel Aviv 6997801, Israel; meitale@tlvmc.gov.il (M.E.); eugenekatchman@gmail.com (E.K.); 2Hadassah Medical Center, Jerusalem 9101002, Israel; mayak@hadassah.org.il (M.K.); oshratisr@gmail.com (O.A.); 3Faculty of Medicine, Hebrew University, Jerusalem 9101002, Israel; yonitw@szmc.org.il (Y.W.-W.); oren.zimhony@weizmann.ac.il (O.Z.); 4Shaare Zedek Medical Center, Jerusalem 9101002, Israel; 5Rambam Medical Center, Haifa 3109601, Israel; y_shahor@rambam.health.gov.il; 6Rappaport Faculty of Medicine, Technion, Haifa 3109601, Israel; regevc@bezeqint.net (R.C.); chazan_b@clalit.org.il (B.C.); sharonre@hymc.gov.il (S.R.); 7Laniado Medical Center, Netanya 4290200, Israel; 8Rabin Medical Medical Center, Petah-Tiqva 4910002, Israel; jihadb@clalit.org.il (J.B.); alaadinat@clalit.org.il (A.A.); 9Faculty of Medicine, Tel Aviv University, Tel Aviv 6997801, Israel; chowersm@post.tau.ac.il (M.C.); yasminm@wmc.gov.il (Y.M.); 10Samson Assuta Ashdod University Hospital, Ashdod 7747629, Israel; talbros@assuta.co.il (T.B.-N.); nirma@assuta.co.il (N.M.); 11Faculty of Health Sciences, Ben Gurion University in the Negev, Beer Sheba 8443944, Israel; nesherke@exchange.bgu.ac.il; 12Soroka Medical Center, Beer Sheba 8443944, Israel; 13Ha’Emek Medical Center, Afula 1834111, Israel; 14Hillel Yaffe Medical Center, Hadera 3810101, Israel; 15Kaplan Medical Center, Rehovot 7610001, Israel; 16Meir Medical Center, Kfar Saba 4428164, Israel; 17Wolfson Medical Center, Holon 5822012, Israel

**Keywords:** *Candida*, *Aspergillus*, COVID-19, critical care, epidemiology

## Abstract

Highly variable estimates of COVID-19-associated fungal diseases (IFDs) have been reported. We aimed to determine the incidence of clinically important fungal diseases in hospitalized COVID-19 patients during the first year of the pandemic. We performed a multicenter survey of IFDs among patients hospitalized with COVID-19 in 13 hospitals in Israel between February 2020 and May 2021. COVID-19-associated pulmonary mold disease (PMD) and invasive candidiasis (IC) were defined using ECMM/ISHAM and EORTC/MSG criteria, respectively. Overall rates of IC and PMD among patients with critical COVID-19 were 10.86 and 10.20 per 1000 admissions, respectively, with significant variability among medical centers. PMD rates were significantly lower in centers where galactomannan was a send-out test versus centers with on-site testing (*p* = 0.035). The 30-day mortality rate was 67.5% for IC and 57.5% for PMD. Treatment with an echinocandin for IC or an extended-spectrum azole for PMD was associated with significantly lower mortality rates (adjusted hazard ratio [95% confidence interval], 0.26 [0.07–0.91] and 0.23 [0.093–0.57], respectively). In this multicenter national survey, variable rates of PMD were associated with on-site galactomannan testing, suggesting under-detection in sites lacking this capacity. COVID-19-related IFDs were associated with high mortality rates, which were reduced with appropriate antifungal therapy.

## 1. Introduction

Invasive fungal diseases (IFDs) have been reported as life-threatening complications of severe viral pneumonia, including cases caused by influenza virus and, more recently, SARS-CoV-2 [1]. The spectrum of fungal diseases in this population includes invasive mold infections, primarily invasive pulmonary aspergillosis [2,3], mucormycosis [4,5], and invasive candidiasis [3]. The epidemiology of COVID-19-associated IFDs varies among geographic regions. Reports of critically-ill COVID-19 patients with invasive aspergillosis have emerged mostly from centers in Europe [2], where influenza-associated pulmonary aspergillosis has also been observed frequently [6]. These reports underscored the difficulties in applying standard IFD criteria to patients with diffuse viral pneumonia and without pre-existing immune deficiency, and led to the formulation of the European Confederation for Medical Mycology (ECMM) and the International Society for Human and Animal Mycology (ISHAM) diagnostic criteria for COVID-19-associated pulmonary aspergillosis (CAPA) [7]. Highly variable estimates of CAPA incidence among critically ill COVID-19 patients have been reported, reflecting differences in diagnostic criteria, patient population, study design, and regional epidemiology [2,3,8,9,10,11,12,13,14,15,16,17]. Mucormycosis was found in 0.29% of hospitalized COVID-19 patients in India [4,18,19], whereas this disease has been rarely reported elsewhere [5].

We undertook a multicenter survey in Israel aimed at assessing the burden of COVID-19-associated fungal diseases during the first three waves of the pandemic (February 2020 to May 2021).

## 2. Materials and Methods

### 2.1. Study Design and Population

This was a multicenter observational cohort study conducted in 13 hospitals in Israel. We collected data on hospitalized patients at least 18 years of age admitted to participating sites with confirmed COVID-19, defined as positive PCR for SARS-CoV-2 from upper or lower airway specimens, between 1 February 2020 and 1 May 2021 and proven or probable IFD (see “definitions”) during the same hospitalization.

The primary study endpoint was the IFD rate, normalized for all COVID-19 admissions and admissions with maximal COVID-19 severity of critical, according to NIH severity definitions [20]. Secondary endpoints included all-cause 30-day mortality, length of hospitalization, length of ICU stay, and duration of mechanical ventilation.

The study was reviewed and approved by the ethics committee of each participating site (approval for the principal site, 0605-15-TLV). Requirement for informed consent was waived considering the retrospective observational nature of the study.

### 2.2. Data Collection and Definitions

Data were retrieved from the electronic medical record system and laboratory computerized database of each center. Collected variables included demographic data, comorbidities (quantified using the Charlson comorbidity score [21]), sequential organ failure assessment (SOFA) score, and treatment of COVID-19, including respiratory and circulatory support, corticosteroids, interleukin-6 inhibitors, low-molecular weight heparin, and remdesivir. Data were collected from each participating center regarding the availability of mycological diagnostics, including fungal culture, galactomannan and PCR (either on-site or as send-out tests), and local policies for performing a mycological work-up of COVID-19 patients.

The ECMM/ISHAM criteria for CAPA were applied generally to pulmonary mold infections [7,22], with some modification of microbiological criteria (Appendix A). Specifically, as proposed by Janssen et al., to account for variability in definitions of non-bronchoscopic lavage across participating centers, we considered bronchial lavage as equivalent to bronchoalveolar lavage [10], and referred to respiratory specimens collectively as airway lavage fluid (ALF). A galactomannan index (GMI) ≥ 1.0, positive culture growing a pathogenic mold or detection of *Aspergillus* species by PCR in either specimen type, serum GMI > 0.5, and positive *Aspergillus* PCR in 2 or more serum or blood samples were considered as mycological evidence [7]. Invasive candidiasis was defined using EORTC/MSG criteria as growth of *Candida* spp. in at least one blood culture or in a normally sterile body site specimen [23]. *Pneumocystis jirovecii* pneumonia was defined based on suggestive radiographic findings and identification of the organism either by dye-based staining or PCR of respiratory specimens. Time to IFD was defined as the time from the first positive SARS-CoV-2 PCR to the date of the specimen that provided microbiological evidence of IFD. Grading of COVID-19 severity was completed according to the NIH severity scale [20].

### 2.3. Statistical Analyses

Data were summarized with descriptive statistics. Between-group comparisons were undertaken using Student’s *t*-test and the Wilcoxon rank sum test for normally and non-normally distributed continuous variables, respectively, and Fisher’s test for categorical variables. Survival analyses were completed using Kaplan Meier plots, and the log-rank test was used to determine the effect of covariates on overall survival. Multivariable survival analyses were undertaken using the Cox proportional hazards method. The proportional hazards assumption was tested for each model by correlating Schoenfeld residuals with time. All calculations were completed in R version 4.1.1 (R foundation).

## 3. Results

COVID-19-related IFDs were surveyed across 13 hospitals. Participating centers included 5 of the 6 tertiary medical centers in Israel, and accounted for 54.9% (1359/2474) and 56.5% (212/375) of total and ICU COVID-19 beds in the country, respectively [24]. Overall, 73 cases of IFDs were identified in 71 patients out of 24,211 COVID-19 hospital admissions between 1 February 2020 and 1 May 2021. IFDs included 40 cases of invasive candidiasis (IC) and 33 cases of pulmonary mold disease (PMD). There were no cases of *Pneumocystis jirovecii* pneumonia. In total, 12 of 13 hospitals (92.3%) reported cases of IC and 9 (69.2%) reported PMD. The overall incidence rates of IC and PMD were 1.65 and 1.36 cases/1000 COVID-19 admissions, respectively, and 10.86 and 10.20 cases per 1000 critical severity COVID-19 admissions, respectively (Figure 1). Significant variability in IFD incidence rates was observed across hospitals, ranging from 0 to 51.2 per 1000 critical hospitalizations for IC and 0 to 37.8 per 1000 critical hospitalizations for PMD (Figure 1). The peak incidence of cases occurred between September 2020 and March 2021, corresponding with the surge of COVID-19 hospitalizations caused predominantly by the Alpha (B.1.1.7) variant (Figure 2).

IFD incidence rates were not significantly different between community and tertiary hospitals (Figure 1). Fungal cultures were performed locally in all centers. Galactomannan was available on-site in 6 (46.1%) of 13 centers, and *Aspergillus* PCR was available in 2 (15.3%). The remaining centers referred these assays to external laboratories. For most centers (10/13, 76.9%), mycological work-up of critical COVID-19 patients was triggered by clinical worsening. Two centers (15.3%) performed periodic screening of airway specimens. The incidence of PMD was significantly higher in centers with on-site galactomannan testing capabilities versus those that utilized off-site laboratories: median (interquartile range [IQR]) 17.4 (8.6–31.5) versus 0 (0–9.14) cases per 1000 critical COVID-19 admissions, respectively; *p* = 0.035). All 4 centers that reported no cases of PMD were community hospitals without local galactomannan testing (Figure 1).

The median age of patients with IFDs was 64 years (IQR, 59–74 years); 47 (66.1%) were males and 24 (33.8%) were females. Baseline clinical features were generally similar between patients with IC and those with PMD (Table 1). Maximal COVID-19 severity was classified as critical in 61 patients (87.1%), severe in 6 (8.5%), and non-severe in 3 (4.2%). In total, 58 (82.8%) patients were hospitalized in the ICU and 61 (87.1%) required mechanical ventilation. Most patients received COVID-19-directed treatments, including corticosteroids (91.3%), remdesivir (33.3%), and tocilizumab (14.4%). A total of 60 patients (86.9%) received treatment with extended-spectrum antibiotics (ceftriaxone, piperacillin-tazobactam, or carbapenems) for suspected or confirmed bacterial infection prior to the diagnosis of IFDs.

### 3.1. Invasive Candidiasis

Invasive candidiasis was diagnosed a median of 19 days (IQR, 12–28 days) after COVID-19 was confirmed, and 16 days (IQR, 8–24 days) after admission to the ICU. Of the 40 patients with IC, 34 (85%) had bloodstream infection, 6 (15%) had *Candida* pleural empyema (2 of whom also had candidemia), and 2 had peritoneal candidiasis (1 with candidemia) (Table 2). Peritoneal candidiasis was due to gastrointestinal perforation in 2 patients with gastric cancer (n = 1) and diverticulitis (n = 1). *Candida* bloodstream infection was attributed to central venous catheters (38.2%, n = 13), pleural empyema (n = 2), and intraabdominal infection (n = 1). The source of bloodstream infection was not determined for 18 patients (52.9%). *Candida glabrata* was the most frequent species (32.5%), followed by *C. albicans* (22.5%) and *C. parapsilosis* (17.5%) (Table 2).

Of 6 patients with pleural empyema, 4 had primary empyema without prior pleural space instrumentation. Two additional patients developed pleural empyema following chest tube insertion for the management of pneumothorax; one of these patients was diagnosed with a bronchopleural fistula. Patients with *Candida* empyema were mechanically ventilated for a median of 24 days before onset of the infection, versus 8 days for patients with other forms of candidiasis (*p* = 0.019). Three patients (50%) with *Candida* empyema were diagnosed with pulmonary embolism versus none of the other patients with invasive candidiasis (*p* = 0.002).

### 3.2. Pulmonary Mold Disease

PMD was diagnosed a median of 16 days (IQR, 10–28 days) after COVID-19 was confirmed, and 10 days (IQR, 3–17 days) after admission to the ICU. Of the 33 patients with PMD, 32 (96.9%) had CAPA, and 1 had pulmonary mucormycosis (Table 3). ALF culture was positive in 26 of 33 patients (78.8%). ALF was tested for galactomannan in 26 patients and was positive in 14 (53.8%). *Aspergillus* PCR was tested on 17 ALF specimens and was positive in 2 (11.8%). Serum galactomannan was tested in 22 patients and was positive in 6 (27.3%). *Aspergillus fumigatus* was the most frequent species identified (n = 13, 39.4%), followed by *A. flavus* (24.2%) and *A. terreus* (15.2%). *Rhizopus arrhizus* was identified in a respiratory specimen of a patient with pulmonary mucormycosis.

Five patients with PMD (15%) had preexisting immune suppression sufficient to be considered host factors according to EORTC/MSG invasive fungal disease criteria [23], including solid organ transplantation (n = 2), active lymphoproliferative malignancy (n = 2), and prolonged use of high-dose corticosteroids (n = 1).

### 3.3. Outcomes

The 30-day mortality rate for all patients with COVID-19-associated IFDs was 63.3% (45/71 patients), with a median survival time of 17 days from the diagnosis of IFD (95% CI, 9 to 28 days; Table 1). The mortality rate was similar for IC and PMD (Figure 3A). Patients spent a median of 31 days in hospital (95% CI 20 to 46 days) and 24 days in the ICU (95% CI, 14 to 37.5 days). The median duration of mechanical ventilation was 22 days (95% CI, 8 to 41 days), and was longer for PMD than for IC (28.5 days versus 13 days, *p* = 0.051; Table 1).

The 30-day mortality rate for patients with IC was 67.5% (27/40), with a median survival time of 8 days (95% CI, 5 to 28 days). Age over 65 years and a SOFA score of greater than 12 were significantly associated with higher mortality rates (Table 4). Patients received primary antifungal treatment with fluconazole (n = 16), an echinocandin (n = 20), and amphotericin B (n = 2). Patients treated with an echinocandin had a significantly lower mortality rate than those treated with fluconazole or amphotericin B (hazard ratio, 0.44, 95% confidence interval [CI] 0.20 to 0.94; *p* = 0.031; Figure 3D). Age, SOFA score, and primary treatment with an echinocandin remained significant predictors of survival in a Cox regression model (Table 4).

The 30-day mortality rate for patients with PMD was 57.5% (19/33), with a median survival time of 24 days (95% CI, 14 to ∞). Patients received antifungal treatment with voriconazole (n = 25), isavuconazole (n = 3), an echinocandin (n = 4), and liposomal amphotericin B (n = 1). Patients treated with an extended-spectrum azole (voriconazole or isavuconazole) had a significantly lower mortality rate than those not treated with either drug (hazard ratio, 0.23, 95% CI 0.093 to 0.57; *p* = 0.00047; Figure 3B). No other variables were associated with survival rate in this patient group.

## 4. Discussion

We report on the incidence of invasive fungal diseases in a large multicenter cohort of hospitalized COVID-19 patients in Israel. The 13 participating hospitals accounted for ~55% of hospital and ICU beds in Israel, and registered 24,211 COVID-19 admissions, including 3037 ICU admissions during the study period. The overall incidence of IFDs was 21 cases per 1000 critical severity COVID-19 admissions, with wide inter-center variability. The incidence of invasive candidiasis was 10.86 cases per 1000 critical severity admissions (range, 0 to 51.2 per 1000 admissions). Previous multicenter studies have shown a somewhat lower incidence of invasive candidiasis in a general ICU population of about 7 cases per 1000 admissions [25,26]; however, invasive candidiasis was more frequent in medical ICUs (19 cases per 1000 admissions) [25]. The incidence of pulmonary mold disease in this cohort was 10.20 cases per 1000 critical severity admissions, with high incidence centers reporting ~30 cases per 1000 admissions, and 4 of 13 centers reporting no cases. Access to galactomannan testing within the hospital appeared to be an important driver of CAPA incidence in this cohort, suggesting that under-diagnosis may be frequent in hospitals lacking this capacity.

Variable estimates of invasive fungal disease incidence among hospitalized patients with COVID-19 have been reported [2,3,4,9,10,11,12,13,14,15,16,17]. Differences in the observed IFD incidence rate may reflect patient selection, local surveillance practices, diagnostic criteria, and possibly the regional epidemiology of airborne mold infections. Centers that employ enhanced mycological screening of mechanically ventilated COVID-19 patients using circulating and respiratory culture and biomarker assays, reported CAPA rates between 10% and 26% [3,9,14]. In contrast, much lower rates of CAPA (0.54%) were observed in a nationwide survey that included 198 hospitals in Japan [11]. Furthermore, while CAPA rates as high as 30% were observed in European ICUs, only 35% of the US Fungal Diagnostic Laboratories Consortium member sites reported recovering *Aspergillus* species from COVID-19 respiratory specimens, and the overall incidence was 2% [15]. A similarly low rate (2%) of invasive mold disease (including CAPA) was observed in a review of COVID-19 autopsies [27].

Most patients with PMD in this cohort did not have predisposing host factors, as defined by the EORTC/MSG [23]. Only 15% had significant pre-existing immune suppression, and 12% had structural lung disease. Prolonged corticosteroid treatment was associated with CAPA in previous studies [22]. Importantly, some of these studies were completed before corticosteroids became the standard of care for critically ill COVID-19 patients. In contrast, almost all patients with PMD in the present cohort (31/33) were treated with dexamethasone for up to 10 days. The effect of corticosteroid treatment during COVID-19 on the risk of PMD has not been fully clarified [24].

The crude 30-day mortality rate was extremely high for patients with invasive candidiasis (67.5%) or pulmonary mold disease (57.5%). Strikingly, only 50% of critically ill patients with invasive candidiasis received primary treatment with an echinocandin, which was significantly associated with greater overall survival. Similarly, treatment of CAPA with extended spectrum azoles was associated with improved overall survival. These findings suggest that efforts to diagnose IFDs and institute early treatment may lead to improved outcomes in this patient population, supporting current treatment guidelines [22].

We aimed to capture the full spectrum of IFDs in COVID-19, including those caused by pathogenic molds, yeasts, and *P. jirovecii*. In addition to IC and CAPA, we found one case of pulmonary mucormycosis and no cases of *P. jirovecii* pneumonia, confirming that these diseases remain uncommon in COVID-19 patients. In total, 6 patients had *Candida* pleural empyema, 4 of whom had no antecedent instrumentation of the pleural space. *Candida* empyema was associated with prolonged mechanical ventilation and pulmonary thromboembolism. A recent review of *Candida* empyema thoracis attributed most cases to thoracic surgery, pleural instrumentation, and esophageal rupture [28], whereas this disease is rare in mechanically ventilated patients without prior pleural manipulation. Our findings suggest that thoracic *Candida* empyema could occur in critically ill COVID-19 patients due to prolonged mechanical ventilation, lung parenchymal necrosis, barotrauma, and overt or occult bronchopleural fistula formation [29,30].

This study was limited by its retrospective, observational design, and lack of standard criteria for diagnosing IFDs among centers. One consequence of this design was the decision to treat alveolar and bronchoalveolar lavage similarly, which may have resulted in overestimation of CAPA rates in some cases. Strengths include a large, multicenter dataset with diverse tertiary and community hospital settings, which allowed us to analyze hospital-related factors associated with IFD rates.

In summary, this multicenter study of critically ill COVID-19 patients in Israel showed a low incidence of pulmonary mold diseases, and rates of invasive candidiasis comparable to those of a general ICU population, with significant variability among centers. Ready access to non-culture diagnostics may be critical to facilitate the diagnosis CAPA. IFDs were associated with extremely high mortality rates, which were reduced in patients who received appropriate antifungal therapy. These findings support recommendations to consider IFDs in clinically deteriorating critically ill COVID-19 patients [22].

## Figures and Tables

**Figure 1 jof-08-00721-f001:**
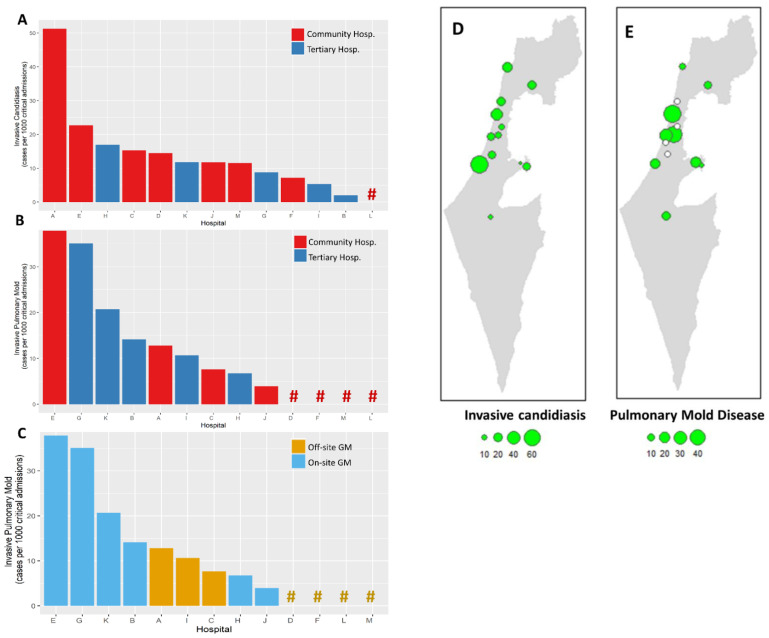
Incidence rates of invasive fungal diseases in 13 hospitals. Incidence rates shown as cases per 1000 hospital admissions of critical severity COVID-19 patients, for invasive candidiasis (**A**) and invasive mold disease (**B**,**C**). Hashtags (#) represent hospitals with 0 incidence rates. Bars and hashtags are colored according to hospital type (**A**,**B**) and local availability of galactomannan testing (**C**). Map of Israel showing the 13 participating centers with incidence rates of invasive candidiasis (**D**) and pulmonary mold disease (**E**). Incidence rate figures represent the number of invasive fungal diseases per 1000 critical COVID-19 admissions. Empty circles represent centers that reported no cases of invasive fungal disease.

**Figure 2 jof-08-00721-f002:**
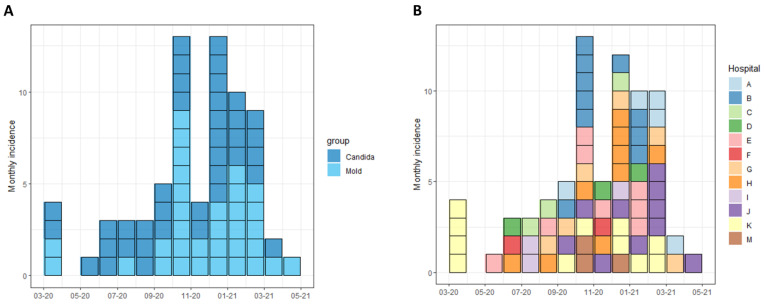
Incidence of invasive fungal diseases in Israeli hospitals, February 2020 to May 2021. Incidence of cases of invasive candidiasis and pulmonary mold disease is shown, by date of hospital admission. Cases are grouped according to clinical syndrome (**A**) and hospital (**B**).

**Figure 3 jof-08-00721-f003:**
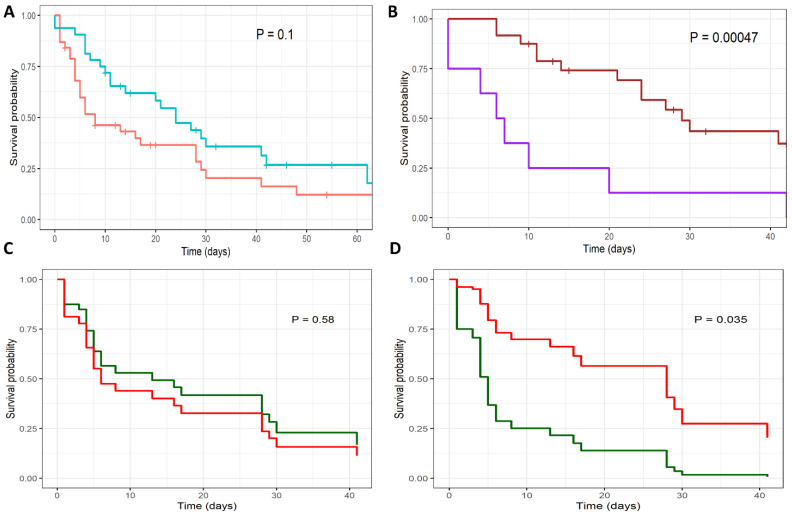
Survival curves of patients with COVID19-associated invasive fungal diseases. Kaplan Meier survival curves are shown for (**A**) patients with invasive candidiasis (red) and pulmonary mold disease (blue); (**B**) pulmonary mold disease, treated with extended-spectrum azole (brown) versus control (purple); Cox regression survival curves are shown for patients with invasive candidiasis (**C**,**D**), comparing treatment effects of (**C**) fluconazole (red) versus control (green); and (**D**) echinocandin (red) versus control (green).

**Table 1 jof-08-00721-t001:** Characteristics of patients with invasive candidiasis and invasive mold disease.

Variable	Invasive Candidiasis	Pulmonary Mold Disease	Total
Total	40	33	71
Sex			
Male	28 (70.0)	21 (63.6)	47 (66.1)
Female	12 (30.0)	12 (36.4)	24 (33.8)
Age, years, median (IQR)	64.5 (57–74.2)	64 (59–74)	64 (59–74)
Charlson Comorbidity score, median (IQR)	2 (1–4)	2 (1–3)	2 (1–4)
COPD	4 (10.3)	4 (12.1)	8 (11.2)
Diabetes mellitus	11 (28.2)	12 (36.4)	24 (33.8)
Ischemic heart disease	9 (23.9)	9 (27.3)	17 (23.9)
Cerebrovascular disease	4 (10.3)	2 (6.1)	6 (8.4)
Immune suppression	6 (15.0)	5 (15.2)	11 (15.1)
Dementia	3 (7.5)	0 (0)	3 (4.2)
LTCF residence	3 (7.5)	0 (0)	3 (4.2)
COVID-19 category			
Non-severe	2 (5.0)	1 (3.0)	3 (4.2)
Severe	5 (12.5)	1 (3.0)	6 (8.4)
Critical	33 (82.5)	31 (93.9)	62 (87.3)
SOFA score, median (IQR)	10 (7.5–13)	7 (5–11)	9 (6–12)
ICU	29 (74.4)	31 (93.9)	58 (81.6)
Mechanical ventilation	32 (82.1)	31 (93.9)	61 (85.9)
Time from diagnosis, median (IQR)	18.5 (12–28)	16 (10–28)	17 (11–28)
MV days before IFD, median (IQR)	12 (3–24)	7 (1–16)	9 (1–18)
ICU days before IFD, median (IQR)	16 (8–24)	10 (3–17)	12 (4.7–18)
CVC	34 (87.2)	28 (84.8)	60 (84.5)
Urinary catheter	33 (84.6)	32 (97.0)	64 (90.1)
Chest tube	6 (15.4)	4 (12.1)	9 (12.6)
Bacterial infection	25 (64.1)	24 (72.7)	46 (64.7)
Corticosteroids	35 (89.7)	31 (93.9)	64 (90.1)
Tocilizumab	5 (12.8)	5 (15.2)	10 (14.0)
Remdesivir	14 (35.9)	9 (27.3)	23 (32.3)
Broad-spectrum antibiotics	32 (82.1)	30 (90.9)	61 (85.9)
Complications			
Pneumothorax/Pneumomediastinum	9 (23.1)	6 (18.2)	13 (18.3)
Venous thromboembolism	3 (7.7)	6 (18.2)	9 (12.6)
Outcomes			
Days in ICU	24 (12–44)	24 (15–35)	24 (14–37.5)
Days on ventilator	13 (3.5–40)	28.5 (15–42)	22 (8–41)
Days in hospital	30 (14–41)	33 (23–52)	31 (20–46)
Crude hospital mortality	27 (67.5)	19 (57.5)	45 (63.3)

Two patients had both invasive mold disease and invasive candidiasis, hence the total column accounts for 71 patients. COPD: chronic obstructive pulmonary disease; LTCF: long-term care facility; SOFA: sequential organ failure assessment; ICU: intensive care unit; CVC: central venous catheter. Broad spectrum antibiotics included carbapenem, piperacillin-tazobactam, and ceftriaxone.

**Table 2 jof-08-00721-t002:** Characteristics of patients with invasive candidiasis.

	BSI	IC	Total
Total	34 (100)	6 (100)	40 (100)
CVC-associated BSI	13 (38.2)	NA	13 (33.3)
Pleural empyema	2 (5.8)	4 (66.6)	6 (15.4)
Peritonitis	1 (2.9)	1 (16.6)	2 (5.1)
Biliary	0 (0)	1 (16.6)	1 (2.6)
Undetermined source	18 (52.9)	NA	18 (45)
*Candida* species			
* C. glabrata*	13 (38.2)	0 (0)	13 (32.5)
* C. albicans*	7 (20.5)	2 (33.3)	9 (22.5)
* C. parapsilosis*	6 (17.6)	1 (16.6)	7 (17.5)
* C. tropicalis*	4 (11.7)	0 (0)	4 (10.0)
* C. lusitaniae*	2 (5.8)	0 (0)	2 (5.0)
* C. krusei*	1 (2.9)	0 (0)	1 (2.5)
* C. dubliniensis*	0 (0)	1 (16.6)	1 (2.5)
* C. auris*	1 (2.9)	0 (0)	1 (2.5)
Mixed species	0 (0)	2 (33.3)	2 (5.0)

BSI: bloodstream infection; IC: invasive candidiasis; CVC: central venous catheter; NA: not applicable.

**Table 3 jof-08-00721-t003:** Microbiological characteristics of pulmonary mold disease.

Variable	N (%)
*Aspergillus species*	
* A. fumigatus*	13 (39.4)
* A. flavus*	8 (24.2)
* A. terreus*	5 (15.2)
* Aspergillus* spp.	6 (18.2)
*Rhizopus arrhizus*	1 (3.03)
Positive Culture	26/33 (78.8)
Positive Galactomannan	
Airway	14/26 (53.8)
Plasma	6/22 (27.3)
Positive PCR	2/17 (11.8)
Total	33 (100)

**Table 4 jof-08-00721-t004:** Survival models for patients with COVID-19-associated IFDs.

Variable	IC	PMD
Age > 65 years	2.84 (1.34–6.02)	1.76 (0.75–4.10)
SOFA score > 12	3.64 (1.46–9.05)	0.76 (0.097–5.99)
Charlson score > 2	1.72 (0.81–3.65)	1.74 (0.70–4.32)
Primary treatment		
Echinocandin	0.44 (0.20–0.94)	1.68 (0.56–5.075)
Fluconazole	0.92 (0.39–2.17)	NA
Extended-spectrum azole	NA	0.23 (0.093–0.57)
Cox proportional hazards model		
Age > 65 years	3.52 (1.26–9.77)	NA
SOFA score > 12	7.37 (2.38–22.77)	NA
Echinocandin	0.26 (0.07–0.91)	NA

SOFA: sequential organ failure assessment; IC: invasive candidiasis; PMD: pulmonary mold disease; NA: not applicable.

## Data Availability

Supporting data are available from the corresponding author upon reasonable request.

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
