# Peer review of "Invasive Fungal Diseases in Hospitalized Patients with COVID-19 in Israel: A Multicenter Cohort Study"

_jof, 2022, doi:10.3390/jof8070721_

Round 1
Reviewer 1 Report
This multicenter study about CAPA incidence in Israel sounds interesting and is well conducted.
I have only a few comments for the authors:
- Dates at lines 64-65 differ from those at l. 71. Please revise.
- At lines 234 and 374: italicize Candida, Aspergillus, respectively.
- Page 11: italicize fungal names.
Reviewer 2 Report
Thank you for the opportunity to review this manuscript intitled “Invasive fungal diseases in hospitalized patients with COVID-19 in Israel: a multicenter cohort study”.
In this paper, the authors conducted an observational multicenter retrospective cohort study of invasive fungal diseases (IFD) among patients hospitalized with COVID-19 in 13 hospitals in Israel between February 2020 and May 2021 to assess the incidence of IFD in severe COVID-19 patients during the first year of the pandemic.
The main findings are:
1) Overall rates of invasive candidiasis and pulmonary mold diseases (mainly invasive pulmonary aspergillosis and a mucormycosis) were 1.65 and 1.36 cases/1000 COVID-19 admissions, respectively, and 10.86 and 10.20 per 1000 admissions, respectively, among patients with critical COVID-19.
2) Pulmonary mold diseases rates were significantly lower in centers where galactomannan was a send-out test versus centers with on-site testing (P=0.035), suggesting under-detection in sites lacking this capacity.
3) High 30-day mortality rates were reported, at 67.5% for invasive candidiasis and 57.5% for pulmonary mold diseases.
4) Treatment with an echinocandin for invasive candidiasis or an extended-spectrum azole for pulmonary mold diseases was associated with significantly lower mortality rates
The authors should be commended on their efforts to provide further insights about this relevant question: what is the incidence of IFD in severe COVID-19? What are the prognostic factors? However, I have the following comments:
1) To considere bronchial lavage as equivalent to bronchoalveolar lavage could led to a overestimation of CAPA diagnosis. Please comment and add a statement in the discussion section (limitations)
2) Please discuss the limitation of the study: retrospective design induce bias in data collection and interpretation…
3) My main concern is about the novelty of the paper. Findings are quite similar to those previously described in numerous studies in the EU.
4) May I ask why the authors limited the inclusion date in 2021? The authors do not provide any input about new variant of concerns, that could be very original and interesting.
One more time, thank you for the opportunity to review your manuscript.
Reviewer 3 Report
In this publication the authors summarise a multicentric Israelian experience in COVID-19 hospitalised patients with IFIs until May 2021. It is well written and easy to follow. However, there are some concerns I would like to raise:
INTRODUCTION:
- Please, include a reference regarding IAPA (influenza-associated pulmonary aspergillosis) in Europe (1st paragraph)
- CAPA incidence: I would suggest to include as reference Salmanton-García Emerg. Infect. Dis. et al and other publications where specific and varied data on incidence are being reported
- For mucormycosis incidence, please cite (Hussain S, Riad A, Singh A, et al. Global Prevalence of COVID-19-Associated Mucormycosis (CAM): Living Systematic Review and Meta-Analysis. J Fungi (Basel) and Hoenigl et al Lancet Microbe)
METHODS:
- Could it be possible to provide data on the participating institutions? Geographic or patient differences, for example? A map?
- Please, use the terms univariaBle and multivariaBle
- Last patient was included more than 1 year ago. Normally, for fungal infections this is not a long time, however, if the authors focus on COVID-19, I would suggest to set the cut-off in a more recent date, if possible.
- I do not really fully understand how were patients classified. ECMM/ISHAM or EORTC/MSG definition was preferred over the other in case of confronting results? May I suggest to include a flow chart explaining how were patients classified?
RESULTS:
- There was no case of IFI due to yeasts other that Candida spp. or these were excluded from analysis?
- Figure 1: I would suggest to use a different colour coding for panel C, as they express different ideas as compared to panels A and B
- How was COVID-19 episode classified as non-severe, severe and critical?
- Table 3: Instead of "undertermined" for Aspergillus spp., I would suggest to Aspergillus spp., or Aspergillus diagnosed through GM, so it can be a bit clearer what you mean
- Figure 3: Could you explain how was the case-control matching for panel B and D performed? I did not read anything about it in methods
- Figure 3: As per figure 1, I would suggest to pay attention to the colour coding, using different colours or patterns depending on the variable to be represented
- I miss information on how were the patients clinically handled (antifungals)
- Patients with with more than 1 IFI, did they have it at the same time?
DISCUSSION:
- Is there any option to comment of IC incidence comparing your data to previous reports? Same for predisposing conditions
- May I suggest to include a paragraph with the limitations you had to face in this research?
- Any reason why sites with outsourced GM had lower incidence rate?
Round 2
Reviewer 2 Report
Thanks to the authors for their responses to the comments.
Once again my main concern is about the novelty and the scientific soundess of this study. Obviously, centers that do not screen for aspergillosis do not diagnose it...
The authors should be commended for conducting a multicenter study. The paper is well organized / presented, so, I defer to the editor's final decision.